# Targeting the Iron-Response Elements of the mRNAs for the Alzheimer’s Amyloid Precursor Protein and Ferritin to Treat Acute Lead and Manganese Neurotoxicity

**DOI:** 10.3390/ijms20040994

**Published:** 2019-02-25

**Authors:** Jack T. Rogers, Ning Xia, Angela Wong, Rachit Bakshi, Catherine M. Cahill

**Affiliations:** 1Neurochemistry Laboratory, Massachusetts General Hospital, Harvard Medical School, Boston, MA 02129, USA; awfy@bu.edu; 2Molecular Neurobiology Laboratory, Massachusetts General Hospital, Harvard Medical School, Boston, MA 02129, USA; NXIA@mgh.harvard.edu (N.X.); RBAKSHI1@mgh.harvard.edu (R.B.)

**Keywords:** translational control, 5’untranslated regions (5’UTRs), Lead/manganese, neurotoxicity, iron, small molecule modulators, APP, ferritin

## Abstract

The therapeutic value of inhibiting translation of the amyloid precursor protein (APP) offers the possibility to reduce neurotoxic amyloid formation, particularly in cases of familial Alzheimer’s disease (AD) caused by APP gene duplications (Dup–APP) and in aging Down syndrome individuals. APP mRNA translation inhibitors such as the anticholinesterase phenserine, and high throughput screened molecules, selectively inhibited the uniquely folded iron-response element (IRE) sequences in the 5’untranslated region (5’UTR) of APP mRNA and this class of drug continues to be tested in a clinical trial as an anti-amyloid treatment for AD. By contrast, in younger age groups, APP expression is not associated with amyloidosis, instead it acts solely as a neuroprotectant while facilitating cellular ferroportin-dependent iron efflux. We have reported that the environmental metallotoxins Lead (Pb) and manganese (Mn) cause neuronal death by interfering with IRE dependent translation of APP and ferritin. The loss of these iron homeostatic neuroprotectants thereby caused an embargo of iron (Fe) export from neurons as associated with excess unstored intracellular iron and the formation of toxic reactive oxidative species (ROS). We propose that APP 5’UTR directed translation activators can be employed therapeutically to protect neurons exposed to high acute Pb and/or Mn exposure. Certainly, high potency APP translation activators, exemplified by the Food and Drug Administration (FDA) pre-approved M1 muscarinic agonist AF102B and high throughput-screened APP 5’UTR translation activators, are available for drug development to treat acute toxicity caused by Pb/Mn exposure to neurons. We conclude that APP translation activators can be predicted to prevent acute metal toxicity to neurons by a mechanism related to the 5’UTR specific yohimbine which binds and targets the canonical IRE RNA stem loop as an H-ferritin translation activator.

## 1. Amyloid Precursor Protein (APP) Translation Inhibitors for Anti-Amyloid Efficacy in Down Syndrome and Familial Alzheimer’s Disease

### 1.1. APP Gene Dose in Down Syndrome and in Rare Cases of Familial Alzheimer’s Disease (fAD)

We review how blocking translation of the mRNA encoding the amyloid precursor protein (APP) using a validated target to decrease its translational expression [1,2]. Conversely, activators of translation of this APP transcript can be used to shield neurons from acute toxicity from heavy metal exposure. The approach to limit APP translation can be viewed as a useful anti-amyloid strategy especially in older Down syndrome (DS) individuals, over age 40, who express three copies of the chromosome 21 gene set (including APP) and who suffer an Alzheimer’s disease (AD)-like dementia confounded often by epileptic seizures [1,3]. Trisomy of the APP gene can be genetically reverted to diploid levels to slow down the features of amyloidosis in DS mouse models [4]. In this section of the article, we outline our approach to inhibit APP expression via the 5’UTR of its transcript. Rare forms of fAD are the result of genetic duplication of the APP gene to increase dose of the Aβ-precursor, as observed in DS [2,5,6] and as associated with so called pathology in “Dup–APP” [7]. In cases of fAD caused by Dup–APP, it has been reported that hemorrhaging is more frequent than in elderly DS subjects [7], suggesting a link between APP and excess release of heme iron that is dampened by other chromosome 21 gene products.

Pertinent to this therapeutic scenario, it is often overlooked that APP facilitates neuroprotective iron export by the universal iron export protein ferroportin [8,9]. Thus, there exists a distinct potential role for APP in neurons to maintain iron homeostasis that has yet to be fully characterized before executing an anti-amyloid strategy. Long et al. (2018) reported a novel upregulation of APP by microRNA-346 (miR-346) *via* targeting of the APP mRNA 5’untranslated region (5’UTR) with implications for iron homeostasis in AD [10]. This paper described the interaction between iron-regulatory protein-1 (IRP1) in conjunction with miR-346 to bind and modulate the activity of the 5’UTR of APP–mRNA with implications for its role when iron interfaces with neural levels of APP and amyloidosis in AD [11]. IRP1 and IRP2 are the recognized RNA binding proteins that control iron homeostasis in the periphery and in terms of cell growth [12,13] and this example of translational regulation includes iron control of APP and ferritin expression in brain neurons [11]. Mounting reports show that APP itself binds to ferroportin to facilitate blood brain barrier iron export as well as iron efflux for from metal burdened neurons [8,14,15,16].

The use of small molecule APP translation blockers will be described in the following sections to list their common properties as therapeutic agents that can remediate the consequences of either too much APP expression as in DS patients (Section 1) [3] or too little APP expression (Section 2). The later likely occurs in acute conditions of environmental exposure to toxic lead (Pb) and manganese (Mn) to affected neurons of exposed children [14,17]. Certainly, triplication (i.e., the presence of three copies) of chromosome 21 in Down syndrome increases APP gene abundance and also the occurrence of clinical amyloidosis and dementia based on the classical amyloid/tau phosphorylation pathway [7]. We would note, based on APP’s iron export function [8], the presence of three copies of APP gene can be predicted to cause excess cellular iron efflux and a local mis-queued anemia in affected cell types, including neurons of the DS individuals [18,19]. In such circumstances of APP over-expression, pharmacological uses of APP translation blockers may provide added clinical efficacy to prevent excess APP/FPN-1 dependent Fe efflux and thus restore iron balance at the same time as generating anti-amyloid efficacy. For example, the drugs phenserine and posiphen, as known APP translation inhibitors, limit Aβ levels in neural cell lines [20] and in mice [1] when administered chronically *in vivo* while their potential effect on intracellular iron homeostasis remains to be assessed [20]. Posiphen is undergoing clinical trials as an APP translation blocker that generates anti-amyloid efficacy [21,22,23,24] and may also cause measurable anemia and would necessitate co-treatment with iron supplements [25].

### 1.2. APP 5’Untranslated Region Translation Blockers as Anti-Amyloid Agents In Vivo

We reported that translational control events generated both by inflammation [26] and iron excess can increase both intracellular APP and Aβ dose by increasing translation via a fully functional iron-responsive element (IRE) RNA stem loop in the 5’UTR of the precursor transcript [11,17,27]. The APP 5’UTR is a uniquely folded RNA structure that is related, though distinct from, the functional IRE in the transcript of the iron storage protein ferritin, though it is base paired in a unique RNA architecture [9,11,17,27]. While there is evidence for a spectrum of IRE like homologies in neurodegenerative disease transcripts [28], the translational control model in Figure 1 summarizes the means by which the APP 5’UTR, while acting in response to iron [11,17], was also sufficiently uniquely folded to act as an anti-amyloid target for the FDA pre-approved anti-cholinesterase phenserine and its enantiomer posiphen [1]. Phenserine and its enantiomer posiphen are each in the class of APP 5’UTR directed drugs that were first identified using transfection-based studies to inhibit APP in cell culture [20] and also limit amyloid Aβ levels *in vivo* [1]. Phenserine and posiphen both were shown to generate anti-amyloid efficacy in primary neurons and in humans (Table 1) [29,30]. The experimental data in Figure 2 recapitulates our previous report that the APP 5’UTR competent transfectants of SH-SY5Y neuroblastoma cells showed anti-APP and anti-amyloid efficacy in response to 100 μM phenserine [20]. Disappointingly, however, phenserine failed phase III clinical trials for AD because of its placebo effect [31]. Nevertheless, the enantiomer posiphen is in current phase II clinical trials as an APP 5’UTR inhibitor that limits levels of brain, serum, and cerebrospinal fluid Aβ amyloid [24] for the purpose of ultimately improving cognition [24,29].

### 1.3. High Throughput Screen for APP 5’UTR Inhibitors for Anti-Amyloid Efficacy

Encouraged by our findings that clinical drugs generated APP 5’UTR directed anti-amyloid activity to provide an *in vivo* validation of this RNA sequence as a therapeutic target, we high throughput screened this same RNA stem loop with non-overlapping collections of 90,000 small molecules from compound libraries at Harvard University [34]. We employed western blot secondary experimental analyses to short-list 13 highly potent 5’UTR-directed APP translation blockers in SH-SY5Y cells (Table 1). All were planar, tri-cyclic molecules, and harbored benzimidazole rings [21]. One of the most active leads was the ninth compound in the series, APP blocker-9 (“JTR-009”) that selectively suppressed APP expression while maintaining alpha-secretase dependent APP(s) secretion from primary cortical neurons as well as in SH-SY5Y cells (Table 1) [2,21].

When considering the anti-amyloid therapeutic use of agents to limit APP translation, we will in the future incorporate into the model the fact that the loss of APP inside the neurons is predicted to reduce the capacity of cells to export iron. Use of any translation inhibitor of APP should result in an embargo of iron export from neurons by the APP–FPN-1 complex. This should indeed reflect cellular exposures to toxic heavy metals such as Pb and Mn which eliminate APP and ferritin from neurons, an event that caused a redox positive embargo of iron export and toxic Fe buildup with accompanying ferroptosis [14,17] (Section 2). This new facet will now be incorporated into experimental protocols when medicinally developing agents such as JTR-009 [21]. This new class of APP translation inhibitors may, in fact, be considered most suitable to treat the orphan disease of APP-trisomy that causes the Dup–APP (hemorrhagic AD) as a form of fAD and also for cases of amyloidosis with complications in DS patients over 40 years of age. Individuals suffering from Dup–APP and DS are associated with increased APP levels and have yet to be investigated for localized anemia in affected neurons. Here, APP 5’UTR directed inhibitors would be expected to potentially restore diploid APP levels, and hence recalibrate cells for loss of Fe, rather than simply to generate anti-amyloid efficacy [7].

## 2. APP 5’UTR Activators Shield Neurons from Acute Pb/Mn Disruption of Fe-Homeostasis

### 2.1. Heavy Metal Neurotoxicity by Loss of IRE/IRP Translation of APP and Ferritin-H Chain

As a rationale, acute, immediate-term Pb toxicity is an exposure risk for children that causes neurodevelopmental delays, autism, and behavioral problems [17,36,37] as accounted in recorded neurological deficits after exposure of children to the metal in Zamfara State, Nigeria [37]. By comparison, occupational exposure to excess manganese (Mn) causes “manganism”, a psychiatric and motor disturbance related to Parkinson’s disease (PD), revealing also perturbed physiological Fe flow from the blood to cerebral spinal fluid [38,39,40].

Like Pb, Mn neurotoxicity was shown to disrupt translation of APP whose intracellular loss prevented acute neuronal iron export by ferroportin (FPN) [9,14,17]. In fact both Pb and Mn interfered with IRE conferred control of translation of APP such that its loss from cells caused an embargo of redox active intracellular iron, an event that appeared to cause neuronal death by ferroptosis [8,14,40,41]. In addition to inhibiting APP/ferroportin levels to embargo iron export, Mn also selectively inhibited translation of L- and H-subunits of ferritin, which is the central neuroprotective iron storage multimer in all cells of the body [9,14,42]. In this section, we will discuss how, while amyloid Aβ accumulation is neurotoxic in aging adults suffering from AD, nevertheless the full-length APP and secreted APP(s) is neurotrophic in younger individuals [17,43,44,45]. APP(s) was shown to combat metal-toxicity to neurons [17]. 

Ferritin (FtH) is the body’s central iron storage protein while APP has a critical role to bind to ferroportin to promote FPN dependent iron export from brain neurons [8,15,16] (Figure 1). We reported that cultured SH-SY5Y neural cells exposed to Pb and Mn exhibited compromised neuronal cell viability, resulting from disruption of iron export and thus increases in the labile iron pool following interference with Fe export and storage as would normally occur for APP and ferritin expression [14,17]. Both metals inhibited translation of ferritin and APP as central iron homeostatic proteins by targeting the IRE RNA stem loops in the 5’UTRs of their mRNAs [14,17]. In sum, there exists a pathway whereby Pb and Mn interfered with ferritin and APP translation *via* their 5’UTR specific IREs to eliminate the presence of these proteins in neurons, sufficient to undermine their iron storage, export, and neuroprotective function [17]. This caused neuronal cell death. Such a toxicity was a primed event generated by an embargo of iron transport and prevention of its storage, an event that increased cellular redox and neuronal death from reactive oxygen species (ROS). These events are known to be accelerated by iron catalysis of hydroxyl radical formation by the Fenton reaction and are associated with ferroptosis [17,46].

### 2.2. Activation of Ferritin and APP Translation to Protect Neurons from Mn/Pb Exposure

To support a therapeutic intervention to prevent Pb and Mn toxicity to neurons, APP over-expression is a proven genetic means to efflux excess embargoed iron and APP over expression indeed provided protection to neurons exposed to these metals [14,17]. For this reason, there is strong rationale for our strategy to pharmacologically induce APP translation and APP(s) secretion to counteract Pb and Mn generated toxicity in vulnerable neurons during increases in toxic metal exposure [35,47].

We propose the therapeutic model shown in Figure 1 whereby defined APP 5’UTR translation activators can be medicinally advanced to protect neurons from short-term metal toxicity in younger Pb and Mn exposed patients, for example in children [37]. This strategy is applicable for patients before an age of increased risk for the onset of amyloidosis. Mouse models demonstrated that pre- and postnatal exposure to Pb is associated with increased risk for amyloidosis and AD specific cognitive deficiencies later in life, though this is not the case at the time of Pb exposure [48,49]. This model has not been tested in Mn exposed rodent models. 

In the next section we will describe how the use of APP translation activators such as the m1 muscarinic acid AF10B are attractive candidates in circumstances while such agents are also co-activators of alpha-secretases, thus eliminating the risk of amyloidosis when pharmacologically increasing the intracellular presence of APP template. There is a strong rationale for the use of small molecules that can restore IRE/IRP dependent APP translation in order to increase intracellular APP and APP(s) as a part of a neuroprotective response to shield neurons from heavy metal neurotoxicity. Any agents that increase APP expression can be predicted to potentially enhance ferroportin activity and to subsequently expel excess embargoed intracellular iron generated in brain neurons after exposure to high doses of Pb and/or Mn.

### 2.3. The Neuroprotective M1 Muscarinic Agonist AF102B, Which Induces Alpha-Secretase, Is an APP 5’UTR Directed Activator

From a chemical viewpoint, (±)-*cis*-2-methyl-spiro(1,3-oxathiolane-5,3′) quinuclidine (AF102B) was a proven M1 agonist that attenuated cognitive dysfunction in AF64A-treated rats [50]. Such M1 muscarinic agonists appear to be disease-modifying agents in AD [51] while cholinergic modulation of APP via the M1 muscarinic receptor has offered a potential therapeutic route to activate alpha-secretase and thus reduce amyloid and increase the secretion of neuroprotective APP(s) [52,53,54,55,56]. Certainly, AF102B is an FDA approved drug currently used for the treatment of Sjogren’s syndrome (an autoimmune disease associated with difficulty breathing, dry mouth, and coughing) [57]. AF102B was shown to generate enhanced neuroprotective APP, likely resulting from alpha-secretase activation of the non-amyloidogenic and neuroprotective pathway of APP expression [51,58]. We reported that this M1 muscarinic agonist confers neuroprotection while inducing APP translation as well as activating alpha-secretase to generate increased secretion of APP(s) from neural cell lines [57].

In Figure 2, we experimentally employed a transfection-based assay to confirm that AF102B indeed increases APP 5’UTR directed expression of a luciferase reporter gene (Figure 2) [57]. SH-SY5Y cells were transfected with a PSV2(APP 5’UTR)–luciferase construct in which the 5’UTR of APP drives luciferase expression. For the purpose of a positive experimental comparison, the anti-cholinesterase phenserine acted as an APP 5’UTR inhibitor in Figure 2 [20]. The data in Figure 1 and Figure 2 shows that AF102B exemplifies a small molecule that activates the APP 5’untranslated region [57] while promoting neuroprotective alpha-secretase expression. This exemplifies a class of drug that activates APP 5’UTR to translate higher levels of APP as a neuroprotectant, a property that may explain its pro-cognitive functions while also generating anti amyloid efficacy via alpha-secretase [53,54,55].

### 2.4. Alternative to Iron Chelation: Novel High Throughput Screened APP Translation Activators in a Mode of Therapeutic Shielding of Neurons from Intracellular Fe Buildup After Pb and Mn Exposure

Based on our findings with AF102B, in contrast to phenserine, we predicted that it would be practical to screen and develop novel activators of APP mRNA translation for the purposes of preventing IRE/IRP dependent loss of this key iron export protein in neurons after exposure to acute high dose Mn and Pb toxicity (Table 2). In fact, small molecule APP 5’UTR directed activators are listed as a result of our high throughput screening campaign conducted at the Columbia University medical center (PUBCHEM AID 1276) and each of these listed agents can theoretically be developed medicinally to restore loss of APP during Pb and Mn exposure. In this event, the iron efflux capacity of cells can be restored to maintain a healthy labile iron pool, which had been lost under conditions where ferritin and APP translation had been blocked after Pb and/or Mn exposure [14]. To sum up, short-term Pb exposure, like that of Mn, was shown to block iron export thereby preventing both APP and ferritin translation. This embargoed iron export increased the labile iron pool (LIP) and caused increased neurotoxic reactive oxygen intermediates (ROS) and neuronal death. Such events may be [17] associated with high levels of acute Pb/Mn toxicity [17]. We propose the use of small molecules screened to activate APP 5’UTR directed translation and can be medicinally developed to improve neuronal survival from Pb and Mn toxicity [17].

Chelation has in the past offered the most direct route to treat Mn and Pb exposed patients [27]. We will list examples of chelation therapy below and also provide a second line of therapeutic intervention which became apparent since we identified a series of small molecule APP 5’UTR activators. These small molecules could offset Pb’s and Mn’s toxic activity as de-stabilizers of neural viability via IRE dependent translational pathways of APP and ferritin translation in prone neurons (Table 2).

Traditionally iron chelators are widely regarded as useful agents to treat conditions of iron overload such as β-thalassemia while having been proven useful for AD [25,65,66] (Table 3). Indeed, there are innumerable advantages supporting the hypothesis for a metal basis for both AD and PD in reference to molecules that serve as scavengers of biometals, such as clioquinol, or molecules that interfere in numerous dysfunctions and pathophysiological changes caused by heavy metals [67,68]. An early milestone study by Crapper-McClachlan et al. (1991) demonstrated the successful clinical use of desferrioxamine (DFO), then deemed an aluminum chelator [66]. Here, intramuscular injection of DFO lead to a significant reduction in the rate of decline of daily living skills in Alzheimer’s disease patients (means (*p* = 0.03) and variances (*p* less than 0.04)) [66]. Elegant studies since then employed intranasal DFO in mice to demonstrate a sparing of affected memory loss, reduced oxidation, and enhanced insulin-signaling in the streptozotocin rat model of Alzheimer’s disease [69]. Deferriprone is a newer line of oral AD treatment that has been tested in clinical trial for PD [70] similar to PBT434 [71] and will be of value to clinically treat AD [72]. The changed metal-protein interaction with iodochlorhydroxyquin (clioquinol) targeted Aβ amyloid deposition and toxicity in Alzheimer disease in a pilot phase II clinical trial [67,68]. Rapid restoration of cognition in Alzheimer’s transgenic mice with 8-hydroxy quinoline analogs is associated with decreased interstitial Aβ [68,73,74,75]. Finally, melatonin is an antioxidant with iron chelation properties that has been in use for sleep disorders for many years [76] and can here be proposed to provide relief in neurodegenerative diseases caused by prion protein mutations that generate fatal familial insomnia [77].

Our collaborative work in the model in Figure 1 showed that forced IRE-independent translation of APP can protect neurons from Pb and Mn toxicity. This model proposes that APP 5’UTR directed activators could generate conditions of intracellular iron efflux from neurons that have been metal (Fe) burdened (Table 2). Of note, in as much as iron surplus is dangerous to neurons, iron deficiency also can alter the expression of genes implicated in AD pathogenesis [18,19]. Iron deficiency in the hippocampus alters the co-operation between neurons in connectomes between developing memory systems; thus, in this system, clearly individual APP inhibitors/activators should be developed to restore intracellular iron balance towards normal homeostatic levels [18,19].

In conclusion, both activators and inhibitors of iron homeostasis can be applied depending on the local conditions of iron in any given clinical circumstance of too much or too little iron. In the case of clinical metal over-exposures, APP and ferritin activators can be developed to re-establish iron homeostasis as accelerated by neuronal ablation from iron overload during the course of metal exposures and even during AD.

### 2.5. Model for APP Translation Activators to Shield Neurons from Pb and Mn Neurotoxicity

We recognize that APP 5’UTR is an mRNA target for translation blockers such as the anti-cholinesterase phenserine and its enantiomer posiphen, each being of sufficient interest to merit clinical trials for their action as anti-amyloid agents in older subjects at risk for AD [24]. By contrast AF102B is a neuroprotective M1 muscarinic agonist in use to treat Sjogren’s syndrome which has been shown to induce APP(s) secretion resulting from alpha-secretase activation of the non-amyloidogenic pathway of APP expression [51,58]. We found AF102B to be an APP 5’UTR dependent translation activator that we propose represents an encouraging starting point for medicinal treatment of acute Pb and Mn toxicity to exposed brain neurons (i.e., in the younger population suffering from environmental/industrial heavy metal exposures long before the onset of an amyloidogenic cascade of AD).

The model in Figure 1 depicts that APP 5’UTR activators such as AF102B can pharmacologically restore APP translation even in the event of Pb and/or Mn interference via APP 5’UTR sequences [10,17]. In this clinical scenario, APP activators such as AF102B and high throughput screened counterparts should be predicted to promote iron efflux by ferroportin when heavy metals generate a redox active iron embargo to affected cells after eliminating neuroprotective APP, APP(s), and ferritin translation [17] (Figure 1 and Figure 2).

## 3. Activators of the Ferritin-H Chain 5’UTR Specific IRE to Shield Neurons from Acute Pb/Mn Disruption of Fe-Homeostasis

The natural product from the indole alkaloid derived from the bark of the *Pausinystalia yohimbe* tree in Central Africa is the source of the antioxidant and anti-arthritic agent commonly known as yohimbine [82]. This natural product is a pre-synaptic alpha 2-adrenergic blocking agent that promotes male potency (PUBCHEM CID 8969). Tibodeau et al. (2005) demonstrated that yohimbine, based on RNA gel shifts, activated translation of ferritin subunits when driven by the IRE RNA stem loop that controls the expression of their transcripts [83] (Table 1). Induction of ferritin subunit translation is a cellular strategy to promote intracellular iron storage to prevent Fe catalyzed redox and ferroptosis while advancing cellular viability and oxidative health in endothelial cells [84]. In the presence of yohimbine, the rate of biosynthesis of ferritin in rabbit reticulocyte lysates was observed to be increased by 40% while this action was supported by a similar increase in synthesis of luciferase protein in a chimera of the IRE and luciferase gene. This identification of yohimbine as a small drug-like molecule that recognizes a naturally occurring IRE in the three-dimensional mRNA structure of ferritin mRNA raises the possibility to develop this agent and others to upregulate ferritin biosynthesis and prevent Fe catalyzed injury thereby preventing ferroptotic toxic injury to neurons following their exposure to heavy metals such as Pb [17] and Mn [14].

We demonstrated that Mn and Pb not only eliminated APP from SH-SY5Y model neurons [17], but that the H-subunit of ferritin was also inhibited by translational mechanisms after exposure of cells to these two toxic metals [14,17]. These actions were also evident in rodent models [14]. It is possible to employ ferritin translation activators such as yohimbine as therapeutic agents to confer protective action in preventing iron overload to neurons affected by Mn and Pb toxic exposure. The natural product yohimbine is a candidate to counter-activate any interference with IRE dependent translation of ferritin L- and H-chains in order to protect affected cells [83].

Related to the action of yohimbine, we identified a novel benzimidazole designated BL-1 from a high throughput screen against the 5’UTR of the prion protein (PrP) transcript (Table 2). BL-1 was originally selected as a means to limit scrapie formation and transmissible encephalopathies (PUBCHEM AID 48862). Nevertheless, PrP had already been shown to regulate iron transport by functioning as a ferro-reductase in human cells [85]. Thus, it was a serendipitous finding when BL-1 showed target selectivity to modulate both PrP and H-ferritin expression (data no shown). To account for this, as well as both proteins conferring iron homeostatic functions, we demonstrated that the 5’untranslated regions in each mRNA demonstrated 78% sequence homology as alternate and unique versions of the IRE that modulates iron dependent translation (Figure 2).

Western blot and cell viability data supports that BL-1 activates expression of the H-subunit of the iron storage protein ferritin which is a known cytoplasmic neurotrophin [84] (Table 3). This is the case also for its mitochondrial variant, mitoferrin [42,86]. Cytosolic ferritin over-expression has protected mice from MPTP (1-methyl-4-phenyl-1,2,3,6 tetrahydropyridine) lesioning [87] while, in Drosophila, forced-expressed of mitochondrial ferritin shortened fly lifespan with behavioral abnormalities [88]. Others have emphasized the protective role of mitochondrial ferritin on erastin-induced ferroptosis [89]. Mitoferrin favorably modulates iron toxicity in a Drosophila model of Friedreich’s ataxia [90]. Like the natural product yohimbine, BL-1 will be further tested to translationally activate IRE dependent translation of ferritin L- and H-chains [83], BL-1 appears to facilitate better iron storage and labile iron pools as a potential new agent to protect neurons, and it will be developed for the clinical treatment of Pb and Mn poisoning.

## 4. Conclusions

Pharmacological lowering of APP levels is appropriate when associated with anti-amyloid efficacy and when excess APP is present in rare trisomy APP mutants of AD [3,97,98]. APP translation blockers are a suitable therapy for forms of AD formed by excess APP in trisomic chromosome 21 dose conditions. Conversely, we discussed that cases of acute high dose short term Pb and Mn poisoning to neurons would benefit from the use of APP activators, sufficient to restore intracellular APP–FPN-1 dependent Fe and metal export and thus improve viability in such affected cells. There is a place for discovering novel agents, capable of adjusting and normalizing brain neuronal APP levels in order to both control amyloid but also to restore and optimize brain iron balance from a toxic iron dys/homeostasis such as during acute Pb/Mn toxicity. Modulation of APP and ferritin translation is a clear means to restore the labile iron pools that have been increased after Pb/Mn exposure has interfered with both ferritin L- and H-chains and APP mRNA translation.

## Figures and Tables

**Figure 1 ijms-20-00994-f001:**
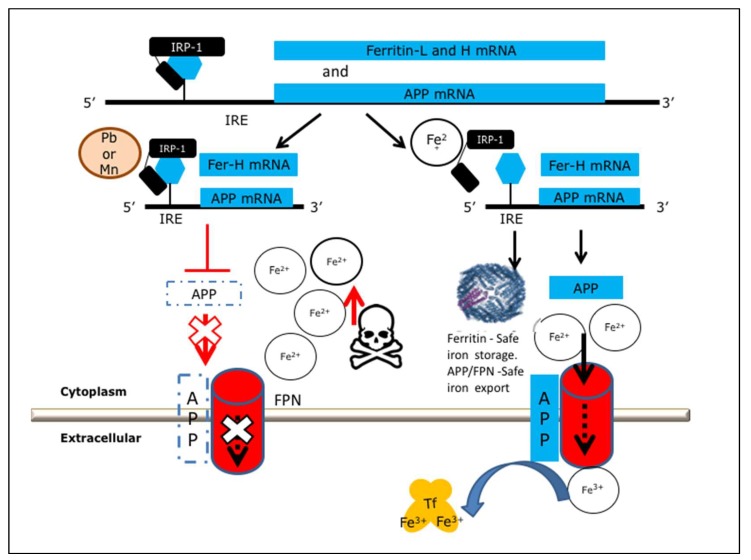
Model for the use of 5’untranslated region directed small molecules as therapeutic agents to offset lead neurotoxicity. These are depicted to act *via* the pathway to activate amyloid precursor protein (APP) and ferritin translation, preventing each of them from maintaining iron homeostasis and iron storage and export functions. Lead blocks translation of APP and thus interferes with ferroportin (FPN)-dependent iron export which normally results in neuroprotective consequences [32]. APP is an iron export protein associated with the onset of Parkinson’s disease with dementia in the event of the loss of tau microtubule associated protein [33]. Adapted from Rogers, et al. 2016, A role for amyloid precursor protein translation to restore iron homeostasis and ameliorate lead (Pb) neurotoxicity. *J. Neurochemistry*, *138*, 479–494 [27].

**Figure 2 ijms-20-00994-f002:**
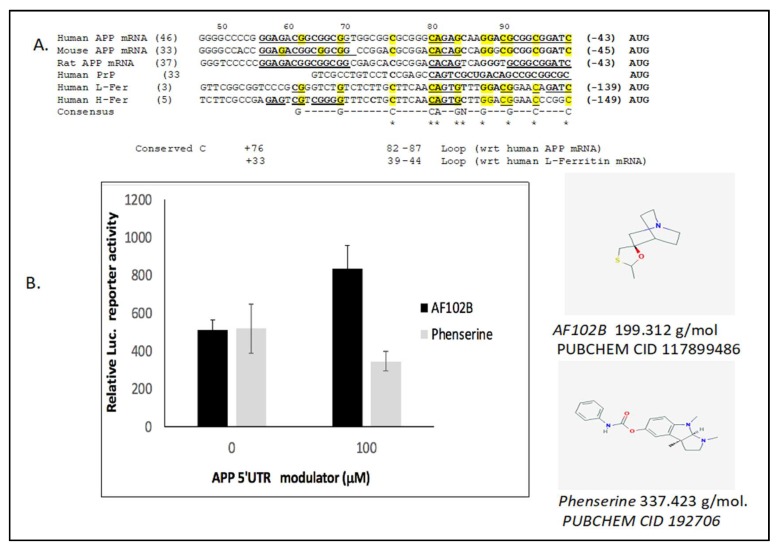
The effect of *AF102B* and phenserine to modulate amyloid precursor protein (APP) conferred expression via the 5’untranslated region (5’UTR) of the APP transcript. Adapted from Rogers et al. (2005) [57]. (**Panel A**) Phylogenetically conserved sequences (underlined; [11]) encoding the 5’UTR APP mRNAs aligned with the CAGUGC H-ferritin IRE sequence that binds to IRP1/IRP2 [59,60]. The homologous CAGAGC motif of the APP iron-response element (IRE) is also in bold lettering and includes the CAGA nucleotides [61], and the AGA tri-loop that also binds IRP [11]. The AGA tri-loop in APP mRNA is comparable to the highlighted AGU tri-loop at the apex of the canonical IRE in the 5’UTRs of ferritin L and H mRNAs [11,59,62,63]. (**Panel B**). The effect of phenserine and AF102B to modulate luciferase reporter gene expression conferred by the APP 5’UTR in pIRES (APP 5’UTR)–luciferase stable SH-SY5Y transfectants (100 µM) [64]. Equal cell numbers (3.9 × 10^5^ cells/well) were treated with APP modulator for 48 h. Significances were calculated using two-way ANOVA followed by Bonferroni post hoc analysis (*N* = 3). Shown: Molecular weights of AF102B and Phenserine (g/mol form PUBSCHEM.

**Table 1 ijms-20-00994-t001:** Small molecules that inhibit APP translation as an anti-amyloid strategy. Therapeutic potential of clinical and high throughput screened (HTS) translation inhibitors of APP mRNA which may be found to impact on amyloid burden in cortical neurons and mouse models of familial Alzheimer’s disease and Down syndrome [21]. Left panel: The anti-cholinesterase phenserine and its enantiomer posiphen are proven APP 5’UTR inhibitors [1]. Right panel: Top high throughput screened (HTS) 5’UTR directed drug APP blocker 9 (JTR-009) inhibited APP translation [21].

APP 5’UTR Inhibitors (In Two Clinical Trials [24,29,31])	HTS APP 5’UTR Inhibitors
Phenserine—Anticholinesterase [31] Enantiomer = Posiphen [1] APP 5’UTR translation blockers [20]. Anti-amyloid. LD-50; 50 µM in SH-SY5Y neuroblastoma cells. Phase-III clinical trials for AD [24,31]. Typical of DUP–APP specific orphan disease drugs to restore APP dose/Fe balance in fAD.	JTR-009 (PUBCHEM AID 1285) was high throughput screened (HTS) in a transfection-based assay to inhibit APP 5’UTR translation conferred to a luciferase reported gene [21,34,35] PubChem AID: 1285 (Available online: https://pubchem.ncbi.nlm.nih.gov/bioassay/1285) JTR-009 can be medicinally optimized to reduce Aβ specific anti-amyloid especially when generated in fAD cases of duplicated over-expression of the APP gene in DS/Dup–APP subjects [21].

**Table 2 ijms-20-00994-t002:** Small molecules that activate APP translation to shield neurons from metal toxicity. Therapeutic potential of both FDA-approved and high throughput screened (HTS) translation activators of APP mRNAs to impact an acute iron overload and associated peroxidative redox stress to heavy metal exposed neurons. **Left Panel:** AF102B is an alpha-secretase activator and neurotrophin clinically employed to treat Sjogren’s disease [78]. This M1 muscarinic agonist was found to also increase APP 5’UTR dependent translation of an APP luciferase reporter gene (Figure 2). **Right Panel:** PUBCHEM Link of Top high throughput screened (HTS) APP 5’UTR directed activators that were identified based on our transfection-based HTSs (Columbia University).

APP 5’UTR Activators (FDA-Approved)	HTS APP 5’UTR Activators
AF102B M1-muscaric agonist [78]. APP(s) and APP 5’UTR translation activator [79]. *N*eurotrophin in the clinic for Sjogren’s disease [80,81]. Rogers et al. [57].	PUBCHEM-AID-1276 (Available online: https://pubchem.ncbi.nlm.nih.gov/bioassay/1276). Agents predicted to promote iron export to rescue neurons from acute Pb- and Mn-induced iron overload and neurotoxicity [14].

**Table 3 ijms-20-00994-t003:** Prion protein/H-ferritin translation modulators to shield neurons from metal toxicity. **A**. Therapeutic modulators of ferritin mRNAs that impact on acute iron overload and associated peroxidative redox stress to neurons. Left panel: Yohimbine is a natural product that increased ferritin 5’UTR dependent translation [83]. Right panel: Iron chelators secondarily reduce the requirement for ferritin translation as an iron storage protein. Molecules that modulate ferritin-H translation adjust iron homeostasis and improve cell-based viability. **B**. BL-1 is a ferritin activator that was screened as an inhibitor of an IRE in the mRNA for the 5’UTR of Prion Protein.

**A. Ferritin IRE Activator**	**H-ferritin IRE Inhibitor**
**Activates ferritin translation.** Yohimbine—Natural product is a neurotrophin that reduces Fe catalyzed oxidative stress. Ferritin-H chain 5’UTR activator, Tibodeau et al. [83].	**Iron chelators inhibit ferritin translation.** Desferrioxamine [91,92]/Deferriprone treat conditions of iron overload and redox-associated stress in neurodegeneration [27,93,94] and cases of iron overload, β-thalassemia transfusion [64]. Melatonin chelates Fe from oxidative/nitrosative damage to cells [76].
**B. HTS Prion Protein 5’UTR Activators**	**HTS Prion Protein Inhibitors**
PUBCHEM AID 1999 (Available online: https://pubchem.ncbi.nlm.nih.gov/bioassay/1999) Predicted to influence Iron REDOX [85,95,96]	PUBCHEM AID 488862 (Available online: https://pubchem.ncbi.nlm.nih.gov/bioassay/488894#section=Top Top PrP 5’UTR blocker is “BL-1” which counter-induced L- and H-ferritin translation to enhance intracellular Fe-storage & cell-based viability.

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
