# Peer review of "Targeting the Iron-Response Elements of the mRNAs for the Alzheimer’s Amyloid Precursor Protein and Ferritin to Treat Acute Lead and Manganese Neurotoxicity"

_ijms, 2019, doi:10.3390/ijms20040994_

Round 1

Reviewer 1 Report

This paper reviews recent knowledge on the transcriptional regulation of amyloid precursor protein (APP) in several pathological conditions. In particular, this study focuses on the therapeutic value of translational inhibition of APP in reducing the neurotoxic amyloid formation especially in some forms of AD formed by excess APP in trisomic chromosome 21 dose conditions. In addition, the Authors discuss that cases of acute high dose short term Pb and Mn poisoning to neurons would benefit from the use of APP activators to sufficient restore of intracellular APP-FPN-1 dependent Fe and metal export and thus improve viability in affected cells.

The work is well written and well organized and cover many aspects of the topic. Also, it covers sufficient background and previous studies in this area to be publishable. I strongly recommend its publication after a minor revision. 

Number of paragraphs: II.2 (page 7, line 234) should be replaced with 2.2

                                    II.3 (page 8, line 273) should be replaced with 2.3

Author Response

We thank the reviewer-! for his/her discerning approval of our paper as outlined in the critique below. We appreciated the support as it is difficult to please everybody, especially these days.

Comments of Ref-1 

The work is well written and well organized and cover many aspects of the topic. Also, it covers sufficient background and previous studies in this area to be publishable. I strongly recommend its publication after a minor revision. 

Number of paragraphs: II.2 (page 7, line 234) should be replaced with 2.2

                                    II.3 (page 8, line 273) should be replaced with 2.3

Response to Ref-1 

We indeed edited the paper a further two times and included the paragraph numbers as you suggested above.

We emailed the completed revision to the editors and will try to upload this changed document as a pdf right now.

Reviewer 2 Report

This review manuscript entitled “Targeting the Iron-responsive Elements of the mRNAs for the Alzheimer’s Amyloid Precursor Protein and Ferritin to treat acute Lead and Manganese Neurotoxicity” analyzed as environmental xenobiotics such as Lead and Manganese induced neuronal cell death through Iron-response element (IRE) sequences of APP and ferritin. Furthermore, they hypothesized that APP 5’untranslated region (5’UTR) directed translation activators may be employed therapeutically to protect neurons exposed to high acute Pb and/or Mn exposure. It is an interesting review that brings extra insight into the new mechanisms of action to counteract the neurotoxic    amyloid formation. The technical performance is correct and the final results are attractive. However,the following points should be considered:

1.    The paper has minor syntax mistakes (which require corrections). For instance, Figure legend 1, page 5, line 169, “these ae depicted”.

2.    The authors should discuss the advantages of their hypothesis in reference to molecules that serve as scavengers of biometals, such as clioquinol, or molecules that interfere in numerous dysfunctions and pathophysiological changescaused by heavy metals, such as melatonin, include a length review article published in Journal of Pineal research (Romero et al. 2014. A review of metal-catalyzed molecular damage: protection by melatonin.J Pineal Res. 2014 May;56(4):343-70) where melatonin has a chelating property which may contribute in reducing metal-induced toxicity.

3.    Figure 2 should be improved. The graphic has several deficiencies. Make uniform the size of numbers

Author Response

 We appreciate the comments of both reviewers and have inserted our accommodated response to alter the text in line with  their editorial comments in the paper (Ref 1 and Ref 2).

 We have incorporated reference to Romero et al., 2014. This paper was indeed a comprehensive review of the uses and biology Melatonin which is an antioxidant and Fe chelator (Ref-2). We included this citation and made this clear and made a suggestion for its development to treat Fatal Familial insomnia.

In section 2 of our paper we accommodated Ref-2’s wish for us to refer also to the metal-based model of Alzheimer’s disease and Parkinson’s disease, which now includes the use of metal chelators such as clioquinol, DFO and deferiprone. There are corresponding citations to match these points (Page 7, Section 2. 2. (iii) (ref 2).

 Figure 2 of this paper had been re-worked to provide statistically significant original APP 5’UTR -luciferase expression data that now is included in an upgraded Figure-2 with the inclusion of consistently sized font.  The figure also reflects the Mol. weights of Phenserine and AF102B and provides their PUBCHEM CID numbers. This figure is currently “in line” with the very valid suggestions of reviewer-2.
